# The Future of Immunotherapy-Based Combination Therapy in Metastatic Renal Cell Carcinoma

**DOI:** 10.3390/cancers12010143

**Published:** 2020-01-07

**Authors:** Rohan Garje, Josiah An, Austin Greco, Raju Kumar Vaddepally, Yousef Zakharia

**Affiliations:** 1Holden Comprehensive Cancer Center, University of Iowa, Iowa City, IA 52242, USA; josiah-an@uiowa.edu (J.A.); yousef-zakharia@uiowa.edu (Y.Z.); 2Department of Internal Medicine, University of Iowa, Iowa City, IA 52242, USA; austin-greco@uiowa.edu; 3Yuma Regional Medical Center, Yuma, AZ 85364, USA; rkvaddepally@gmail.com

**Keywords:** renal cell carcinoma, checkpoint inhibitors, VEGF inhibitors, mTOR inhibitors

## Abstract

In the past two decades, there has been a significant improvement in the understanding of the molecular pathogenesis of Renal Cell Carcinoma (RCC). These insights in the biological pathways have resulted in the development of multiple agents targeting vascular endothelial growth factor (VEGF), as well as inhibitors of the mammalian target of the rapamycin (mTOR) pathway. Most recently, checkpoint inhibitors were shown to have excellent clinical efficacy. Although the patients are living longer, durable complete responses are rarely seen. Historically, high dose interleukin 2 (IL2) therapy has produced durable complete responses in 5% to 8% highly selected patients—albeit with significant toxicity. A durable complete response is a surrogate for a long-term response in the modern era of targeted therapy and checkpoint immunotherapy. Numerous clinical trials are currently exploring the combination of immunotherapy with various targeted therapeutic agents to develop therapies with a higher complete response rate with acceptable toxicity. in this study, we provide a comprehensive review of multiple reported and ongoing clinical trials evaluating the combination of PD-1/PD-L1 inhibitors with either ipilimumab (a cytotoxic T-lymphocyte-associated protein 4, CTLA-4 inhibitor) or with anti-VEGF targeted therapy.

## 1. Introduction

Kidney cancer is the second most common malignancy arising from the urinary system. In 2019, about 73,820 new cases and 14,770 deaths were estimated to occur in the United States [1]. According to the SEER (Surveillance, Epidemiology, and End Results) program, about 16% patients with RCC present with distant metastatic disease and have a 5-year survival rate of 11.6%. Clear cell is the most common histological subtype of renal cell carcinoma (ccRCC) and accounts for about 75% of the cases. The remaining 25% non-clear cell subtypes include papillary, chromophobe, medullary, collecting duct, sarcomatoid, and unclassified types. Advanced, unresectable, or relapsed RCC is treated with palliative systemic therapy.

In the past two decades, there has been significant progress in the understanding of the molecular pathogenesis of ccRCC. These insights in the biological pathways have resulted in the development of multiple agents targeting vascular endothelial growth factors (VEGF) such as sunitinib, pazopanib, cabozantinib, axitinib, and lenvatinib, as well as inhibitors of the mammalian target of rapamycin pathway (mTOR) such as temsirolimus and everolimus. These agents have improved the objective response rates (ORR) and overall survival (OS); however, they are short-lived. The ability of the tumor cells to evade the host immune surveillance by PD-1-PD-L1 interaction, lead to the development of checkpoint inhibitors that target either PD-1 or PD-L1, and restore immune competence. Nivolumab, a fully human IgG4 programmed death 1 (PD-1) checkpoint inhibitor selectively blocks the interaction between PD-1 (expressed on activated T cells) and PD ligands (expressed on immune cells and tumor cells). In a Phase III trial of patients previously treated with one or two VEGF inhibitors, nivolumab, when compared to everolimus, showed significantly longer overall survival along with fewer adverse events [2]. Despite the addition of several agents to the therapeutic armamentarium of ccRCC, durable complete responses are rarely seen. Historically, high dose interleukin 2 (IL2) has achieved durable complete responses, but only in 5% to 8% of highly selected patients, and at the cost of significant toxicities [3,4]. Patients who withstand this toxic therapy and achieve complete response can essentially be cured. This points to the fact that achieving a complete response is a surrogate for long term response in the modern era of targeted therapy and checkpoint immunotherapy. Newer treatment strategies that aim to achieve complete response similar to IL-2, albeit with less toxicity, is a plausible therapeutic avenue. One such approach is to explore the synergy of checkpoint inhibitors with various targeted therapeutic agents, which, if successful, is a step closer to achieve potential cure.

## 2. The Rationale to Combine Immunotherapy with Angiogenesis Inhibitors

Vascular endothelial growth factors (VEGF) play a crucial role in tumor angiogenesis by binding to VEGF receptors. In addition to a proangiogenic effect, they also exert immunosuppression in the tumor microenvironment by not only inducing the accumulation of myeloid-derived suppressor cells and regulatory T cells but also by impeding the migration of T lymphocytes towards the tumor microenvironment. VEFG inhibition can restore antitumor immunity by normalizing the vasculature and endothelial cell activation [5]. Additionally, PD-1 blockage also promotes cytotoxic T-cell infiltration into the tumor and significantly enhances antitumor immunity. The synergy of both VEGF inhibition and PD-1 blockade was successfully demonstrated in murine cancer models [6]. The clinical evidence and correlative biomarkers of synergism were initially reported in a Phase II clinical trial, where atezolizumab, a PD-L1 inhibitor, in combination with bevacizumab, was compared to sunitinib in patients with treatment naïve metastatic ccRCC. This study evaluated molecular biomarkers predicting response by utilizing the genes signatures associated angiogenesis (Angio^High^ or Angio^low^), anti-tumor immune response (T_eff_^High^ or T_eff_^Low^), and myeloid inflammation (Myeloid^High^ or Myeloid^Low^) [7,8,9,10]. In tumors with high angiogenesis, these were no progression-free survival (PFS) differences with either sunitinib monotherapy or the combination of atezolizumab and bevacizumab. On the contrary, in the Angio^low^ subgroup, the combination therapy had better PFS when compared to sunitinib. Additionally, the high T_eff_ gene signature expression, a maker of a high preexisting anti-tumor immune response, was associated with improved PFS with atezolizumab  +  bevacizumab when compared to sunitinib monotherapy. A higher level of PD-L1 expression by immunohistochemistry was also predictive of a better response to the combination therapy. However, tumor mutation and neoantigen burden did not correlate with PFS [9,10]. Finally, higher expression of myeloid inflammation gene signatures (associated with impaired antitumor T cell response), did not show any difference between the combination therapy versus sunitinib [9,10]. This study—with its correlative biomarkers—shows evidence of the potential synergy of immune checkpoint inhibitors and VEGF inhibitors.

Numerous of these combinational studies are ongoing, and some have been recently published. In this review, we summarize the clinical trials evaluating the combination of PD1/PDL1 inhibitors with either ipilimumab (a cytotoxic T-lymphocyte-associated protein 4, CTLA-4 inhibitor) or angiogenesis inhibitors.

## 3. Nivolumab in Combination with Ipilimumab versus Sunitinib Monotherapy

In Checkmate 214, a Phase III randomized open-label multicenter trial, nivolumab (3 mg/kg) + ipilimumab (1 mg/kg) was compared with sunitinib monotherapy (50 mg daily for 4 weeks on and 2 weeks off every cycle) in patients with treatment naïve metastatic ccRCC [11] (summarized in Table 1). Of the 1096 enrolled patients, 550 received nivolumab + ipilimumab, and 546 received sunitinib; 425 and 422, respectively, had intermediate or poor-risk disease as per the International Metastatic Renal-Cell Carcinoma Database Consortium (IMDC) prognostic model [12]. At a median follow-up of 25.2 months, the 18-month overall survival of the intermediate and poor-risk patients was 75% with nivolumab + ipilimumab and 60% with sunitinib. The median OS was not reached with the combination therapy vs. 26.0 months with sunitinib (hazard ratio for death, 0.63; *p* < 0.001). Progression-free survival (PFS) and overall response rates (ORR) also favored the checkpoint inhibitors when compared to sunitinib and were 11.6 months vs. 8.4 months and 42% vs. 27% (*p* < 0.001), respectively. The complete response (CR) rate was 9% in the combination immunotherapy arm. However, the PFS and ORR were better with sunitinib monotherapy in patients with IMDC favorable risk cancer. Additionally, PDL-L1 status was not predictive of response to the combination therapy. Treatment-related grade 3 or 4 adverse events (AE) occurred in 250 (46%) and 335 (63%) patients in nivolumab + ipilimumab and sunitinib groups, respectively. The most common grade 3 or 4 AEs in the combination group were elevated lipase levels, fatigue, and diarrhea. While in the sunitinib group, the most common grade 3 or 4 AEs were hypertension, fatigue, palmar-plantar erythrodysesthesia, and elevated lipase levels. About 35% of patients in the combination immunotherapy group required high-dose steroids for the management of immune-mediated adverse events. There were eight treatment-related deaths in the combination group and four in the sunitinib group. Based on the study results, the US Food and Drug Administration (FDA) approved the combination immunotherapy for intermediate and poor-risk patients in the first-line setting for metastatic ccRCC and also received a category 1 recommendation by the National Comprehensive Cancer Network (NCCN). Additionally, Grunwald and colleagues studied the depth of response as an indicator for long term survival among the 1096 patients in Checkmate 214 with previously untreated ccRCC [13]. They found that patients who received nivolumab + ipilimumab had similar OS between >50–≤75% and >75% tumor reduction. Receiver operating characteristic analysis was utilized to show that >50% depth of reduction indicated the most OS benefit in nivolumab + ipilimumab. This study showed that nivolumab + ipilimumab treatment resulted in prolonged OS in comparison to sunitinib, and depth of response may reflect the possibility of long-term survival for ccRCC patients who receive nivolumab + ipilimumab [13].

## 4. Pembrolizumab in Combination with Axitinib in Metastatic ccRCC

In Phase III, randomized KEYNOTE-426 clinical trial of the efficacy of checkpoint PD-1 inhibitor, pembrolizumab (200 mg IV every 3 weeks) in combination with axitinib (5 mg orally twice daily) was compared to sunitinib monotherapy in previously untreated patients with metastatic ccRCC [14]. In this study, 861 patients were randomized to receive treatment in one of these two groups until disease progression or intolerable toxicities. The median PFS was both clinically and statistically significant with the pembrolizumab—axitinib group when compared to sunitinib (15.1 months vs. 11.1 months, HR 0.69; *p* < 0.001) irrespective of IMDC risk groups and PDL1 status. The median OS was not reached, but the risk of death was 47% lower with combination therapy when compared to sunitinib. This OS benefit was also evidenced in all subgroups irrespective of age, metastatic sites, and IMDC risk groups. The ORRs were better with the combination therapy when compared to sunitinib and were 59.3% vs. 35.7%, respectively. The most common grade 3 and 4 treatment-related adverse events in both groups were diarrhea and hypertension. The incidence of hepatic toxicity was higher in the pembrolizumab-axitinib group; however, there were no deaths related to hepatoxicity. Based on the significant efficacy and acceptable toxicity profile, this combination therapy was approved by the FDA for treatment naïve metastatic ccRCC, irrespective of PDL1 status or IMDC risk stratification. Brian Rini and colleagues presented a subgroup analysis of KEYNOTE-426 during the American Society of Clinical Oncology (ASCO) annual meeting in June 2019, examining intermediate and poor-risk groups in addition to the sarcomatoid group [17]. Analysis of 592 patients with intermediate and poor-risk mCCRCC patients showed 1 year OS, median PFS, ORR, and complete response rate at 87.3%, 12.6 months, 55.8%, and 4.8% in the pembrolizumab + axitinib group compared to the sunitinib group at 71.3%, 8.2 months, 29.5%, and 0.7%. In addition, 105 patients with sarcomatoid features were examined and results showed improved 12 months OS, PFS, ORR, and complete response rate at 83.4%, median PFS not reached, 58.8%, 11.8% in pembrolizumab + axitinib group while sunitinib group showed 79.5%, 8.4 months, 31.5%, and 0% [17]. This study confirmed the benefit of pembrolizumab + axitinib intermediate and poor IMDC risk groups and in tumors with sarcomatoid features.

## 5. Avelumab in Combination with Axitinib in Metastatic ccRCC

TK Choueiri and colleagues evaluated the combination of avelumab, a PD-L1 antibody with axitinib in a Phase 1b clinical trial, where it was not only shown to be safe, but also had 58% objective response rate in patients with metastatic ccRCC [18]. These encouraging results lead to Phase III, randomized clinical trial (JAVELIN Renal 101), which evaluated the efficacy of avelumab (10 mg/kg IV every 2 weeks) along with axitinib (5 mg twice a day) and compared to sunitinib in the first-line setting for metastatic ccRCC. Of the 886 patients, 442 were randomly assigned to the combination group, and 444 were assigned to receive sunitinib; 560 (63.2%) patients had PD-L1 positive tumor defined as ≥1% of immune cells staining positive within the tumor area with Ventana PD-L1 (SP263) assay [15]. The median PFS was significantly longer with the combination therapy when compared to sunitinib and was 13.8 vs. 8.4 months in the overall population and 13.8 vs. 7.2 months in PD-L1 positive patients, respectively. Additionally, irrespective of PD-L1 status, the objective response rates and complete response rates were almost doubled with combination therapy as compared to sunitinib and were 51.4% vs. 25.7% and 3.4% vs. 1.8%, respectively [15]. On subgroup analysis of PDL1 positive patients, irrespective of IMDC prognostic risk group the combination therapy was better than sunitinib in terms of median PFS and objective response rates. Interestingly, patients did better for those who underwent nephrectomy in the combination group, but no difference was noted if patients did not undergo nephrectomy. However, only a small fraction of patients do not undergo nephrectomy in this subgroup. In terms of safety, grade 3 or higher adverse events were seen in 71.2% and 71.5% in the combination therapy and sunitinib groups, respectively. About 166 patients (38.2%) who received avelumab and axitinib had immune-mediated adverse events, and about 11% required high dose glucocorticoids. The data was premature for overall survival analysis and a further follow up is needed [15]. Another subgroup analysis of the JAVELIN Renal 101 trial presented at the European Society for Medical Oncology (ESMO) annual meeting held in September 2019 included 117 patients with advanced ccRCC who did not undergo upfront cytoreductive nephrectomy and had renal lesions [19]. In this study, Albiges and colleagues found that 34.5% of the patients that received avelumab and axitinib had ≥30% shrinkage in renal lesions in comparison to 9.6% who received sunitinib, indicating future areas of research in neoadjuvant therapy with immunotherapy and tyrosine kinase inhibitor [19].

## 6. Atezolizumab in Combination with Bevacizumab versus Sunitinib Monotherapy in Metastatic ccRCC

In a Phase II randomized study (IMmotion150) reported by McDermott et al., atezolizumab monotherapy or in combination with bevacizumab was compared with sunitinib in 305 patients with treatment naïve metastatic renal cell carcinoma. In the intention to treat population, the median PFS was 11.7 months vs. 6.1 months vs. 8.4 months (HR = 1.00; 95% CI = 0.69–1.45) with atezolizumab plus bevacizumab (*n* = 101), atezolizumab alone (*n* = 103) and sunitinib (*n* = 101), respectively, and was not statistically significantly. In PD-L1 positive patients (defined as ≥1% PD-L1 expression on IC by IHC), the atezolizumab plus bevacizumab arm had a PFS of 14.7 months as opposed to 7.8 months with sunitinib. Treatment-related grade 3 or 4 adverse events were seen in 40% vs. 17% vs. 57% in atezolizumab plus bevacizumab, atezolizumab, alone, and sunitinib arms, respectively. In the atezolizumab plus bevacizumab group, proteinuria was the most common adverse event leading to treatment discontinuation. While in the atezolizumab monotherapy group, it was nephritis, pancreatitis, and demyelination. In the sunitinib group, there was increased blood creatinine and palmar-plantar erythrodysesthesia syndrome. Of note, there were two treatment-related AEs leading to death in the sunitinib group secondary to sudden death and intestinal hemorrhage, and one in the atezolizumab plus bevacizumab group secondary to intracranial hemorrhage [9].

The above findings were further supported in the interim results of IMmotion151, a randomized Phase III trial comparing atezolizumab plus bevacizumab combined therapy vs. Sunitinib in treatment naïve patients with metastatic renal cell carcinoma [20]. OS was immature at the time of interim analysis and results for this were not reported. PFS in intention-to-treat patients was greater in the atezolizumab plus bevacizumab group at 11.2 months as compared to 8.4 months in the sunitinib group. Of note, this PFS benefit was shown across analyzed subgroups including MSKCC risk, liver metastases, and sarcomatoid histology. Also, in patients who had disease with positive PD-LI status (characterized as >/=1%) PFS for atezolizumab plus bevacizumab group was longer than in the sunitinib group at 11.2 months and 7.7 months, respectively. Grade 3 and 4 AEs occurred in 40% of the atezolizumab plus bevacizumab group and 54% of the sunitinib group. Discontinuation of therapy secondary to all grade AEs occurred in 12% and 8% for the two groups, respectively. Subgroup analysis of IMmotion151 examined 142 patients with sarcomatoid histology and found that median PFS, median OS, ORR, and complete response rate were 8.3 months, median OS not reached, 49%, and 10% in atezolizumab + bevacizumab group compared to sunitinib group with 5.3 months, 15 months, 14%, and 3%. MD Anderson Symptom Inventory (MDASI) scale analysis indicated median time to deterioration was 11.3 months in atezolizumab + bevacizumab group compared to 4.9 months in sunitinib group. In addition, biomarker analysis with angiogenesis signature subset and T-effector gene expression subset were higher in sarcomatoid histology when compared to non-sarcomatoid tumors, while PD-L1 positive tumors were seen in greater proportion in sarcomatoid histology at 63% compared to 39%. This study showed that biomarker analysis was consistent with the improved survival seen in atezolizumab + bevacizumab group compared to sunitinib group in tumors with sarcomatoid histology [17]. Table 1 provides a summary of all the studies with results.

In addition to the above Phase III trials, there are multiple other combination therapies with preliminary Phase I results and ongoing Phase III studies.

## 7. Nivolumab in Combination with Tivozanib in mRCC

In an open-labeled non-randomized Phase Ib/II study, tivozanib was studied in combination with nivolumab in patients with metastatic renal cell carcinoma and previously treated with one oral TKI [21]. Tivozanib is a highly selective VEGF receptor tyrosine kinase inhibitor with minimal off-target action and presumably lower AE profile. In the dose-escalation phase, six patients were enrolled, and no dose-limiting toxicities (DLTs) were observed. The maximum tolerated dose (MTD) was found to be full dose tivozanib, 1.5 mg/day, in combination with nivolumab. In the Phase II cohort, an additional 22 patients were enrolled, and grade 3 and 4 AEs were seen in 60% of patients, with hypertension being the most common. The objective response rate was 56%, with one patient achieving complete response [22]. Median PFS was 18.5 months for patients who were previously untreated and did not reach PFS for patients who were previously treated at the time of analysis [23]. ORR was 56% and results were premature for OS analysis.

## 8. Nivolumab in Combination with Sunitinib or Pazopanib in mRCC

CheckMate 016 study was a Phase I trial dose-escalation and expansion study, which evaluated the safety and efficacy of nivolumab, in combination with either sunitinib or pazopanib in patients with metastatic renal cell carcinoma [24].

Twenty patients were enrolled in the nivolumab plus pazopanib group. However, 4 DLTs were seen with this combination, including elevated liver enzymes (*n* = 3) and fatigue (*n* = 1). Also, 70% (14/20) patients experienced grade 3 and 4 treatment-related AEs. This study arm was subsequently closed due to significant toxicities. The objective response rate was 45%, and all were partial responses. The median PFS and OS were 7.2 months and 27.9 months, respectively.

In the nivolumab plus sunitinib cohort, 33 patients were enrolled. There were no DLTs. In this cohort as well, grade 3 and 4 treatment-related AEs were seen in 82% (27/33) patients, and the most common were hypertension, hepatotoxicity, diarrhea, and fatigue. The objective response rate was 54.5%, with two complete responses and 16 partial responses. The median PFS was 12.7 months, and median OS was not reached. Though there were signals for antitumor efficacy, due to significant toxicities, this combination was also not considered for further evaluation [24].

## 9. Pembrolizumab in Combination with Bevacizumab in mRCC

In the Big Ten Cancer Research Consortium sponsored Phase Ib/II clinical trial, the safety and antitumor activity of pembrolizumab in combination with bevacizumab was evaluated in 61 patients with metastatic ccRCC, who at received at least one prior systemic therapy [25]. No dose-limiting toxicities had been reported. The 200 mg fixed dose of pembrolizumab and 15 mg/kg dose of bevacizumab, both given every three weeks, was determined to be safe and recommended for a multicenter Phase 2 study. The overall response rate was 60.9% with median time on treatment of 298 days and median PFS was found to be 17.0 months. The most common grade 3 or 4 AEs were hypertension, proteinuria, and adrenal insufficiency. One death was reported due to heart failure [25].

## 10. Pembrolizumab in Combination with Lenvatinib in mRCC

Lenvatinib is an oral multi kinase inhibitor that targets VEGFR1–3, FGFR1–4, PDGFRα, and the oncogenes *RET* and *KIT* [26]. A Phase Ib/II multicenter open-label study of lenvatinib plus pembrolizumab in patients with clear cell metastatic RCC was performed to assess the safety and antitumor activity [27]. Patients with measurable metastatic RCC with or without prior systemic therapy with ECOG status ≤1 were enrolled. Lenvatinib 20 mg daily plus pembrolizumab 200 mg intravenously every three weeks was deemed as the maximum tolerated dose and recommended for Phase 2 evaluation. Of the 33 patients enrolled, the objective response rate was 52% with disease control rate of 94% [28]. Median follow up time for PFS was 4.2 months. Median PFS was not reached, but the PFS rate at 3, 6, and 9 months were 93.1%, 73.8%, and 64.6%, respectively. Response duration of ≥6 months was seen in 80.8% of patients. The most common therapy-related adverse events were fatigue, followed by dysphonia, diarrhea, stomatitis, hypertension, dry mouth, nausea, proteinuria, and hand-foot syndrome. Treatment was discontinued in 9% of patients due to adverse events [28].

## 11. Pembrolizumab and Pazopanib in Patients with Advanced or mRCC

Preliminary safety and antitumor effects of the Phase I part of the KEYNOTE-018 study were presented at the ASCO annual meeting 2017 [29]. Twenty patients were enrolled in two cohorts to study safety in pembrolizumab and pazopanib combination therapy and determine the maximum tolerated dose. Cohort A evaluated pembrolizumab 2 mg/kg every three weeks and pazopanib 800 mg daily. Cohort B evaluated pembrolizumab at the same dose and pazopanib at 600 mg daily. Seven out of 20 patients experienced DLTs primarily in the form of hepatotoxicity. Subsequently, 15 patients were added to Cohort C, where pazopanib was started with nine weeks run-in, followed by combination therapy to limit toxicities. In the initial five patients enrolled, three had DLTs that included pneumonitis, bowel perforation and grade 4 lipase levels. Due to significant DLTs and grade 3 and 4 adverse events, the combination of pazopanib and pembrolizumab has overall limited tolerability and was not suitable for evaluation in a larger cohort [29].

## 12. Cabozantinib in Combination with Atezolizumab in Advance Renal Cell Carcinoma

COSMIC-021 clinical trial is a Phase Ib, multicenter trial that evaluated the safety and efficacy of the combination of cabozantinib (VEGFR/MET/AXL inhibitor) at 40 mg or 60 mg daily dose along with atezolizumab 1200 mg every three weeks in seven genitourinary cohorts, including a cohort of advanced RCC. There were no DLTs but the 40 mg cabozantinib dose was chosen for the expansion cohort. Of the 10 evaluable patients, the ORR was 50% with 1 CR, 4 PR. The most common grade 3 AEs were hypertension, diarrhea and hypophosphatemia [30].

Several other studies investigating other combinations of antiangiogenic agents and immunotherapies in mRCC are ongoing. Ongoing trials include VEGF/PD-1 blockade combination, including lenvatinib with everolimus and pembrolizumab, and nivolumab with cabozantinib as first-line treatment in patients with mRCC are summarized in Table 2 with their current accrual status.

## 13. Discussion

The therapeutic landscape of renal cell carcinoma has rapidly advanced in the past decade. The combination strategies of checkpoint inhibitors along with ipilimumab or antiangiogenic agents have demonstrated good synergy. Most notably, for the IMDC intermediate and poor-risk patients, the dual checkpoint inhibitor combination of nivolumab with ipilimumab and the combination of pembrolizumab with axitinib have shown improved response and overall survival leading to FDA approval in the first-line setting. Similarly, for the IMDC favorable-risk, the combination of pembrolizumab with axitinib or single-agent VEGF inhibitors such as sunitinib and pazopanib are preferred first-line options [31]. The combination of avelumab and axitinib is also an alternative option in the first-line setting based on improved response rate and PFS, but there was no OS benefit at the time of data cut off [15]. Figure 1 illustrates the first-line treatment approach for mRCC. The benefit of the combination therapy was seen in a reasonable number of patients irrespective of any specific biomarkers, such as PD-L1 positivity. Also, the complete response rates were higher from the combination strategies, whether this translates to long term survival or potentially a cure is unknown. Long-term follow-up data from these studies are needed.

It is imperative to understand that not all patients benefit from combination therapy, and some have a higher incidence of serious toxicities. Biomarkers and genomic studies that can predict the efficacy of either monotherapy or combination therapy should be explored to individualize the treatment. A few combinations appeared to be beneficial in limited cases with specific histology and immunologic biomarkers and may be considered in particular situations, as seen in atezolizumab and bevacizumab in comparison to sunitinib monotherapy, where PFS was more significant in sarcomatoid variant mRCC and PD-L1 expressing tumors [10]. Not all combination therapies are safe. The combination of pazopanib with either nivolumab or pembrolizumab was associated with significant hepatoxicity and not deemed safe for further study. Even the combination of axitinib and pembrolizumab had high grade 3 and 4 elevations of hepatic enzymes. The exact mechanism of such toxicity is unknown. Caution should be maintained before initiating such combination therapies in patients with liver dysfunction at baseline.

Combination therapy can be associated with severe side effects, with a significant overlap in the toxicity profile of the drugs. For example, diarrhea and hepatoxicity are common side effects that can be associated with either axitinib or checkpoint inhibitors. The management varies based on the etiology and also the severity, i.e., treatment discontinuation/dose reduction and use of loperamide for diarrhea in case of axitinib and timely initiation of high dose corticosteroids for autoimmune colitis and hepatitis secondary to checkpoint inhibitors. Checkpoint inhibitor and antiangiogenic combination therapy requires intense monitoring on treatment that entails frequent office visits, physician and nurse assessments, investment of health-care resources, and higher financial burden on the patient. A multidisciplinary approach and an in-depth knowledge of complications are crucial before embarking on the combination therapy.

## 14. Conclusions

In summary, the combination of checkpoint inhibitors and anti-angiogenic agents is a valuable addition in the therapeutic armamentarium for mRCC. The choice of combination therapy or monotherapy should be individualized after a thorough discussion between the physician and patients based on cost, efficacy, and toxicity profile.

## Figures and Tables

**Figure 1 cancers-12-00143-f001:**
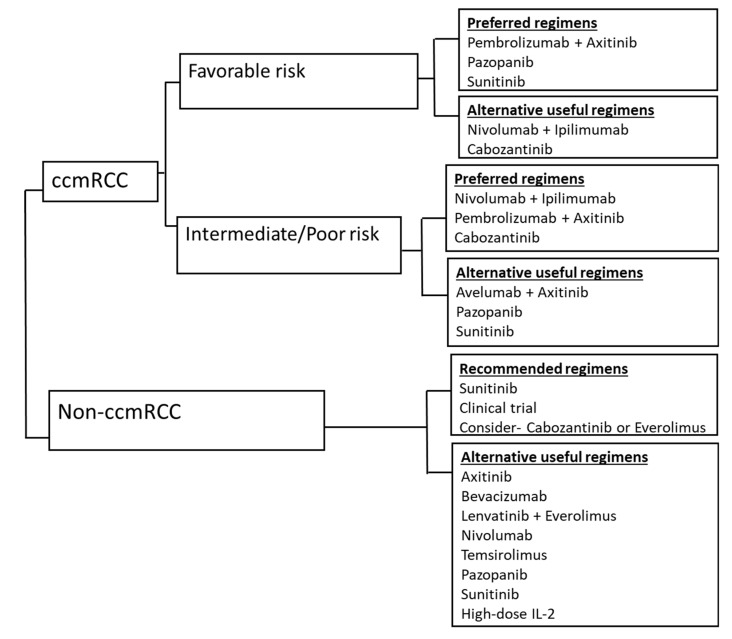
Overview of treatment strategy for treatment naïve metastatic RCC, based on IMDC risk-stratification. ccmRCC, clear cell metastatic renal cell carcinoma; non-ccmRCC, non-clear cell metastatic renal cell carcinoma; risk stratification based on International Metastatic Renal Cell Carcinoma Database Consortium (IMDC) criteria. The International Metastatic RCC Database Consortium (IMDC) prognostic model integrates six adverse factors: Karnofsky performance status (KPS) <80 percent, time from diagnosis to treatment < 1-year, hemoglobin concentration < lower limit of normal, serum calcium > upper limit of normal, neutrophil count > upper limit of normal, platelet count > upper limit of normal. (Favorable risk: no risk factors, intermediate risk: 1 or 2 risk factors, poor risk: 3 or more risk factors).

**Table 1 cancers-12-00143-t001:** Immunotherapy based combination trials in treatment-naive mRCC with results.

Study	*N*	Compounds	Median OS, mo (95% CI)	Median PFS, mo (95% CI)	CRR	ORR (95% CI)	Grade 3 and 4 TRAEs	Treatment-Related Deaths	Treatment Discontinuation Rate	IRAE Needing ≥40 mg Total Daily Dose of Prednisone or Equivalent
Checkmate 214 [11]	1096	**Intermediate and poor risk:** Nivolumab + ipilimumab vs. sunitinib	NR vs. 26.0 HR = 0.63; *p* < 0.001.	11.6 vs. 8.4 (HR = 0.82; *p* = 0.0331.	9% vs. 1%	42% vs. 27%	46% vs. 63%	1.5% vs. 0.74%	22% vs. 12%	35%
KEYNOTE-426 [14]	861	Pembrolizumab + axitinib vs. sunitinib	NR, HR 0.53; *p* < 0.0001 12-mo OS: 90% vs. 78%	15.1 vs. 11.1 HR 0.69; 0.57–0.84; *p* = 0.0001)	5.8% vs. 1.9%	59.3% vs. 35.7%; *p* < 0.0001	62.9% vs. 58.1%	0.9% vs. 1.6%	both drugs: 30.5%, sunitinib: 13.9%	N/a
JAVELIN Renal 101 [15]	886	Avelumab plus axitinib vs. sunitinib	NR; 12-mo: 86% vs. 83% (HR 0.78; 0.55 to 1.08; *p* = 0.14)	13.8 vs 8.4 (HR 0.69; 0.56 to 0.84; *p* < 0.0001)	3.4% vs 1.8%	51.4% vs. 25.7 %	71.2% vs. 71.5%	0.7% vs. 0.2%	7.6 vs.13.4	11.1%
IMmotion151 [16]	915; PDL1+: 362	Atezolizumab + bevacizumab vs. sunitinib	NR, 24-mo: 63% vs. 60% (HR 0.93; 0.76 to 1.14; *p* = 0.4751)	ITT: 11.2 vs. 8.4 (HR 0.83; 0.70–0.97; *p* = 0.0219) PDL1+: 11.2 vs. 7.7	ITT: 5% vs. 2%; PD-L1+: 9% vs. 4%	ITT: 37% vs. 33% PD-L1+: 43% vs. 35%	40% vs. 54%	1.1% vs. 0.22%	5% vs. 8%	9%

OS, overall survival; CI, confidence interval; PFS, progression-free survival; ORR, objective response rate; CRR, complete response rate; NR, not reached; N/a, not available; HR, hazard ratio; mo, months; TRAEs, treatment-related adverse events; IRAE, immune-related adverse events.

**Table 2 cancers-12-00143-t002:** Select ongoing VEGF/PD-1 blockade combination studies in mRCC.

National Clinical Trial ID Number (Study)	Treatment	Phase	Clinical Trial Status	Treatment Line	Patients	Primary Outcome Measures
**Lenvatinib**						
NCT02811861 (CLEAR)	Lenvatinib with everolimus or pembrolizumab compared to SOC sunitinib	III, randomized 1:1:1, open label	active, not-recruiting	First-line treatment of subjects with advanced renal cell carcinoma	1069	PFS by independent review
**Cabozantinib**						
NCT03141177 (CheckMate 9ER)	Nivolumab + Cabozantinib vs. Sunitinib	Phase III, randomized, open-label study	active, not recruiting	First line, metastatic RCC	638	PFS per blinded independent central review (BICR)
NCT03937219 (COSMIC-313)	Cabozantinib + nivolumab + ipilimumab vs. Nivolumab + ipilimumab	III, randomized, 1:1, double-blind	recruiting	First line, intermediate- or poor-risk metastatic RCC	676	PFS by blinded independent radiology committee (BIRC)
NCT03793166 (PDIGREE study)	Nivolumab and Ipilimumab followed by Nivolumab vs. Cabozantinib with Nivolumab	III, randomized, open label	recruiting	First line	1046	OS
NCT03149822	Pembrolizumab plus Cabozantinib	I/II, open label, single arm	recruiting	First or second line	55	ORR (CR + PR)
NCT03200587	Avelumab and Cabozantinib	Ib, open label	recruiting	First line	20	DLTs, AEs, RP2D

OS, overall survival; PFS, progression-free survival; ORR, objective response rate; CR, complete response; PR partial response; DLT, dose limiting toxicity; AE, adverse event; RP2D, recommended Phase 2 dose.

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
