# Peer review of "The Future of Immunotherapy-Based Combination Therapy in Metastatic Renal Cell Carcinoma"

_cancers, 2020, doi:10.3390/cancers12010143_

Round 1

Reviewer 1 Report

In the present manuscript the authors review some clinical trials evaluating the combination of certain checkpoint inhibitors with angiogenesis inhibitors to treat patients with metastatic renal cell carcinoma. The reported data provide evidence that treatment with some combinations can result in improved survival in comparison to, for example, sunitinib monotherapy. However, some combinations were associated with severe toxicity and they did not lead to improved survival. Therefore, what makes a particular combination therapy work on a group of patients?

With respect to metastatic renal cell carcinoma and many other cancers, the selection of first-line agents or combination therapies is primary based on comparisons of clinical data as reviewed in this manuscript or the experiences of individual physicians. Genetic and/or immunologic biomarkers that are predictive of clinical responses are important. Hence, the auhors should address this issue thoroughly to decide which mono and combination therapires will be of highest utility in the treatment of metastatic renal cell carcinoma.

A graphical overview of various treatment strategies for metastatic renal cell carcinoma would be useful.

Author Response

We thank you for the valuable feedback as this was helpful to indentify the limitations of the manuscript. we will have several changes (with track review on).

In the present manuscript the authors review some clinical trials evaluating the combination of certain checkpoint inhibitors with angiogenesis inhibitors to treat patients with metastatic renal cell carcinoma. The reported data provide evidence that treatment with some combinations can result in improved survival in comparison to, for example, sunitinib monotherapy. However, some combinations were associated with severe toxicity and they did not lead to improved survival. Therefore, what makes a particular combination therapy work on a group of patients?

we have added couple of paragraphs in the discussion about monotherapy vs combination therapy. currently, there are no well defined biomarkers for treatment selection for patients.

With respect to metastatic renal cell carcinoma and many other cancers, the selection of first-line agents or combination therapies is primary based on comparisons of clinical data as reviewed in this manuscript or the experiences of individual physicians. Genetic and/or immunologic biomarkers that are predictive of clinical responses are important. Hence, the auhors should address this issue thoroughly to decide which mono and combination therapires will be of highest utility in the treatment of metastatic renal cell carcinoma.

we have added we few lines to address this concerns.

A graphical overview of various treatment strategies for metastatic renal cell carcinoma would be useful.

this was added.

we are open for further suggestions

Reviewer 2 Report

The manuscript by Garje and coworkers is a very good written review about an immune-based combination therapy in renal cell carcinoma. Authors have described in a very precise and detailed way treatment plans based on various approaches, which might be of high interest for many MD searching for a summary of available therapies or searching for alternative therapy. Further, authors have included a very clear and helpful table summarize their review. Except for a very informative structure of the review, authors have combined up-to-date knowledge about the regulation of signaling pathways in RCC and possible treatment strategies. Overall, it is a very good review, sufficiently loaded with valuable information and a new perspective on immunotherapy in RCC.

Minor concerns:

Use the term: “immunotherapy-based” (with a hyphen) in the title and continue throughout the whole manuscript to gain more clarity for the readers.

Author Response

thanks for your positive feedback, we made changes as per your suggestion